# Using the Krylov Subspace Formulation to Improve Regularisation and Interpretation in Partial Least Squares Regression

## Abstract

Partial least squares regression (PLS-R) has been an important regression method in the life sciences and many other fields for decades. However, PLS-R is typically solved using an algorithmic approach, rather than through an optimisation formulation and procedure. There is a clear optimisation formulation of the PLS-R problem based on a Krylov subspace formulation, but it is only rarely considered. The popularity of PLS-R is attributed to the ability to interpret the data through the model components, but the model components are not available when solving the PLS-R problem using the Krylov subspace formulation. We therefore highlight a simple reformulation of the PLS-R problem using the Krylov subspace formulation as a promising modelling framework for PLS-R, and illustrate one of the main benefits of this reformulation—namely that it allows arbitrary penalty terms of the regression coefficients to be included in the PLS-R model. Further, we propose an approach to estimate the PLS-R model components for the solution found through the Krylov subspace formulation, that are those we would have obtained had we been able to use the common algorithms for estimating the PLS-R model. We illustrate the utility of the proposed method on simulated and real data.

## 1 Introduction

Partial least squares regression (PLS-R) was proposed by Wold et al. (1983) as an alternative to principal component regression (PCR, Massy, 1965) and ridge regression (Hoerl & Kennard, 1970) for the problem of approximating concentrations of constituents in a chemical sample from spectroscopic data of the sample.

With spectroscopic data, the variables are typically strongly correlated and numerical instability problems may therefore arise when using ordinary least squares regression. PLS-R solves this by projecting the data onto an orthogonal set of basis vectors, derived from the data itself, and linear regression is then performed within this subspace. This is very similar to how PCR works, but instead of using the singular value decomposition (SVD) of the data, PLS-R is based on the partial least squares path modelling method in mode A (Wold et al., 1983), where the linear subspace is derived using both the data and the target variables. This leads to a non-linear shrinkage estimator (Krämer, 2007), with some unusual properties (Butler & Denham, 2000).

Nevertheless, PLS-R has been widely used and is a very common choice of regression method in the chemical and biological literature (Frank & Friedman, 1993; Wold et al., 2001; Boulesteix & Strimmer, 2007; Gromski et al., 2015), but also in *e.g.* neuroimaging (Krishnana et al., 2011) and many other fields. PLS-R has been a popular method within these fields not only for regression, but extensions have also been developed for classification (Sjöström et al., 1986; Ståhle & Wold, 1987; Barker & Rayens, 2003), outlier detection (Valderrama et al., 2007), and even for unsupervised problems such as *e.g.* clustering (Kloss et al., 2015). One of the reasons for the popularity of PLS-R is likely because of the possibility to interpret the resulting regression vector in terms of the basis vectors of the solution subspace, the PLS-R score and loading vectors give information about linear relationships between samples, but also about which variables correlate with those

relationships. This is of course an important aspect, especially considering the recent growth in the field of interpretable and explainable machine learning (Linardatos et al., 2020). The fact that PLS-R allowed better interpretations to be made, compared to PCR and ridge regression at the time, is likely to have added to the popularity of the PLS-R method.

PLS-R has also been extended in many other ways, for instance to handle multiple target variables (Bisani et al., 1983; Mateos-Aparicio, 2011). There are also several similar but different formulations of the underlying optimisation problem, that make their own trade-offs and typically give different but related or similar solutions (Helland, 1988; de Jong, 1993). There are also different forms of preconditioning or preprocessing methods, that are sometimes equivalent to the PLS-R problem, but allow different interpretations of the resulting subspace (*e.g.*, Trygg & Wold, 2002; Ergon, 2005; Kvalheim et al., 2009).

Regularised versions of PLS-R have also been proposed, such as sparse PLS-R (Lê Cao et al., 2008), elastic-net PLS-R (Chun & Keleş, 2010), and non-negative PLS-R (Allen et al., 2013). However, those methods regularise the score and/or loading vectors, which means that the resulting regression vector need not be sparse.

While it is possible to solve the PLS-R optimisation problem using any general purpose solver, the most common ones appear to be to use the SVD, or the non-linear iterative partial least squares (NIPALS) algorithm—an instance of the power method (Abdi, 2010). While there are also accelerated versions of the power method (Xu et al., 2018; Rabbani et al., 2021), the original NIPALS algorithm (Wold et al., 1983) appears to still be one of the most common solvers for the PLS-R problem.

It has been shown that the PLS-R regression vector lies in a Krylov subspace (Helland, 1988; Rosipal & Krämer, 2006; Krämer, 2007), and through that has connections to both Lanczos bidiagonalization (Eldén, 2004) and the conjugate gradient method (Wold et al., 1984). However, these relations do not seem to be exploited in practice. A reason for this could be that while it is easy to solve the problem using the Krylov subspace formulation, it is not immediately possible to obtain the scores and loadings from the obtained regression vector, reducing the model interpretability.

In this work, we present solutions to two main problems: First, we show how to use the Krylov subspace formulation to allow general-purpose regularisation terms to be added to the PLS-R problem. In particular, we analyse a regularised version of the Krylov formulation of the PLS-R problem that results in a sparse regression vector. The regularisation we used was the elastic net, *i.e.* a linear combination of $\ell_1$ and squared $\ell_2$ penalties (Zou & Hastie, 2005). Second, we propose a means to estimate the score and loading vectors for the found regression vector that we would have gotten, had we been able to use the traditional NIPALS solver for the regularised PLS-R problem. This procedure thus allows the same interpretation of the model, data, and target values as in classical PLS-R but now also when using the Krylov subspace formulation.

## 2 Method

### 2.1 Partial Least Squares Regression

We consider the standard linear regression problem, *i.e.*,

$$y_i = \mathbf{x}_i^{\mathrm{T}} \mathbf{w} + \varepsilon_i,$$

for $i = 1, \ldots, n$, where $y_i \in \mathbb{R}$ is a continuous target variable, $\mathbf{x}_i \in \mathbb{R}^p$ is a data sample of $p$ measured variables, and $\varepsilon_i \sim \mathcal{N}(0, \sigma^2)$ is zero-mean additive Gaussian noise with variance $\sigma^2$. The data samples, $\mathbf{x}_i$, and target variables, $y_i$, are assumed to be zero-mean.

In this setting, the PLS-R problem is typically formulated as (Höskuldsson, 1988),

$$\underset{\mathbf{w}_1 \in \mathbb{R}^p}{\text{maximise}} \ \mathbf{y}^{\mathrm{T}} \mathbf{X}_0 \mathbf{w}_1 \tag{1}$$
$$\text{subject to} \ \|\mathbf{w}_1\|_2^2 = 1,$$

where $\mathbf{y} \in \mathbb{R}^n$ is the vector of all target variables (all vectors are assumed to be column vectors), $\mathbf{X}_0 := \mathbf{X} \in \mathbb{R}^{n \times p}$ contains the $n$ data samples in the rows, and $\mathbf{w}_1 \in \mathbb{R}^p$ is a weight vector. Once the weight vector

is found, a score vector is computed as $\mathbf{t}_1 = \mathbf{X}_0 \mathbf{w}_1$ and a loading vector as $\mathbf{p}_1 = \mathbf{X}_0^{\mathrm{T}} \mathbf{t}_1/(\mathbf{t}_1^{\mathrm{T}} \mathbf{t}_1)$. We also compute a $\mathbf{y}$-loading, $c_1 = \mathbf{y}^{\mathrm{T}} \mathbf{t}_1/(\mathbf{t}_1^{\mathrm{T}} \mathbf{t}_1)$. Once all score and loading vectors are found, the data matrix, $\mathbf{X}_0$, is *deflated*, by anti-projecting on the found score vector,

$$\mathbf{X}_1 = \mathbf{X}_0 - \mathbf{t}_1 \mathbf{p}_1^{\mathrm{T}} = \mathbf{X}_0 - \frac{\mathbf{t}_1 \mathbf{t}_1^{\mathrm{T}} \mathbf{X}_0}{\mathbf{t}_1^{\mathrm{T}} \mathbf{t}_1} = \left( \mathbf{I} - \frac{\mathbf{t}_1 \mathbf{t}_1^{\mathrm{T}}}{\mathbf{t}_1^{\mathrm{T}} \mathbf{t}_1} \right) \mathbf{X}_0.$$

After deflation, the optimisation program in Equation 1 is run again, using $\mathbf{X}_1$, to find a second set of weights, $\mathbf{w}_2$, scores, $\mathbf{t}_2$, and loadings, $\mathbf{p}_2$ and $c_2$. A sequence of $K \leq \mathrm{rank}(\mathbf{X}) \leq \min(n, p)$ such vectors are thus constructed, and we collect them as the columns in the matrices

$$\mathbf{W} = [\mathbf{w}_1, \ldots, \mathbf{w}_K], \quad \mathbf{T} = [\mathbf{t}_1, \ldots, \mathbf{t}_K],$$

$$\mathbf{P} = [\mathbf{p}_1, \ldots, \mathbf{p}_K], \quad \text{and} \quad \mathbf{C} = [c_1, \ldots, c_K].$$

This procedure leads to mutually orthogonal weight and score vectors. A final regression vector is computed as

$$\boldsymbol{\beta}_{\mathrm{PLS}} = \mathbf{W}(\mathbf{P}^{\mathrm{T}} \mathbf{W})^{-1}(\mathbf{T}^{\mathrm{T}} \mathbf{T})^{-1} \mathbf{T}^{\mathrm{T}} \mathbf{y} = \mathbf{W}(\mathbf{P}^{\mathrm{T}} \mathbf{W})^{-1} \mathbf{C}^{\mathrm{T}}, \tag{2}$$

and new samples are predicted as

$$\hat{y}_{\mathrm{new}} = \mathbf{x}_{\mathrm{new}}^{\mathrm{T}} \boldsymbol{\beta}_{\mathrm{PLS}}.$$

The PLS-R method is thus a complicated procedure, and the steps leading to Equation 2 are fairly opaque, and typically in need of careful individual study to fully understand. Further, the deflation procedure is sensitive to numerical precision in the solution, and any errors are propagated to higher order components (Björck & Indahl, 2017). It would be very difficult to include regularisation terms in Equation 1, that penalises the regression vector, $\boldsymbol{\beta}_{\mathrm{PLS}}$, (through Equation 2) since the problem would become highly non-linear and a very complicated function of the weight vectors, $\mathbf{w}$, and especially so with more elaborate regularisers. In the next section, we present an alternative, but equivalent formulation of the PLS-R problem in which we can trivially incorporate penalties of the regression vector.

## 2.2 Partial Least Squares and Krylov Subspaces

Helland (1988) showed that an alternative basis for the weight vectors is the sequence, $\mathbf{X}^{\mathrm{T}} \mathbf{y}, (\mathbf{X}^{\mathrm{T}} \mathbf{X}) \mathbf{X}^{\mathrm{T}} \mathbf{y}$, $\ldots, (\mathbf{X}^{\mathrm{T}} \mathbf{X})^{K-1} \mathbf{X}^{\mathrm{T}} \mathbf{y}$, generating a Krylov subspace (Watkins, 2007). We have the following definition.

**Definition 1.** *A Krylov subspace of order $K$, generated by a matrix $\mathbf{A} \in \mathbb{R}^{m \times m}$ and a vector $\mathbf{v} \in \mathbb{R}^m$, is the linear subspace spanned by the first $K$ powers of $\mathbf{A}$, and is denoted by*

$$\mathcal{K}_K(\mathbf{A}, \mathbf{v}) = \mathrm{span}\{\mathbf{v}, \mathbf{A}\mathbf{v}, \mathbf{A}^2 \mathbf{v}, \ldots, \mathbf{A}^{K-1} \mathbf{v}\}.$$

Now, since the PLS-R weight vectors all lie in $\mathrm{span}\{\mathbf{X}^{\mathrm{T}} \mathbf{y}, (\mathbf{X}^{\mathrm{T}} \mathbf{X}) \mathbf{X}^{\mathrm{T}} \mathbf{y}, \ldots, (\mathbf{X}^{\mathrm{T}} \mathbf{X})^{K-1} \mathbf{X}^{\mathrm{T}} \mathbf{y}\}$ (Helland, 1988), we immediately obtain the following result. This result is known, but we have failed to find a direct proof of it; we therefore provide a simple proof, for completeness.

**Lemma 1.** *If $\mathcal{K}_K(\mathbf{X}^{\mathrm{T}} \mathbf{X}, \mathbf{X}^{\mathrm{T}} \mathbf{y})$ is a basis for the weight vectors, $\mathbf{W} = [\mathbf{w}_1, \ldots, \mathbf{w}_K]$, then the PLS-R regression vector, $\boldsymbol{\beta}_{\mathrm{PLS}}$, lie in the Krylov subspace of order $K$ generated by $\mathbf{X}^{\mathrm{T}} \mathbf{X}$ and $\mathbf{X}^{\mathrm{T}} \mathbf{y}$, i.e.*

$$\boldsymbol{\beta}_{\mathrm{PLS}} \in \mathcal{K}_K(\mathbf{X}^{\mathrm{T}} \mathbf{X}, \mathbf{X}^{\mathrm{T}} \mathbf{y}).$$

*Proof.* It is well-known that the Krylov subspace $\mathcal{K}_K(\mathbf{X}^{\mathrm{T}} \mathbf{X}, \mathbf{X}^{\mathrm{T}} \mathbf{y})$ is a basis for the weight vectors (see *e.g.*, Helland, 1988). Hence, if we let $\mathbf{K} \in \mathbb{R}^{p \times K}$ be some basis for $\mathcal{K}_K(\mathbf{X}^{\mathrm{T}} \mathbf{X}, \mathbf{X}^{\mathrm{T}} \mathbf{y})$, then

$$\mathbf{W} = \mathbf{K}\mathbf{A},$$

for some matrix $\mathbf{A} \in \mathbb{R}^{K \times K}$. From Equation 2, we have

$$\boldsymbol{\beta}_{\mathrm{PLS}} = \mathbf{W}(\mathbf{P}^{\mathrm{T}} \mathbf{W})^{-1}(\mathbf{T}^{\mathrm{T}} \mathbf{T})^{-1} \mathbf{T}^{\mathrm{T}} \mathbf{y} = \mathbf{W}\mathbf{v},$$

with $\mathbf{v} = (\mathbf{P}^T\mathbf{W})^{-1}(\mathbf{T}^T\mathbf{T})^{-1}\mathbf{T}^T\mathbf{y}$. Thus

$$\boldsymbol{\beta}_{\text{PLS}} = \mathbf{W}\mathbf{v} = \mathbf{K}\mathbf{A}\mathbf{v} = \mathbf{K}\boldsymbol{\alpha},$$

where $\boldsymbol{\alpha} = \mathbf{A}\mathbf{v}$ is a vector. This concludes the proof. $\qquad\square$

Hence, we see that the PLS-R problem can be cast in the form of a linear least squares problem, as

$$\underset{\boldsymbol{\beta}\in\mathbb{R}^p}{\text{minimise}} \quad \frac{1}{2n}\|\mathbf{y} - \mathbf{X}\boldsymbol{\beta}\|_2^2$$
$$\text{subject to} \quad \boldsymbol{\beta} \in \mathcal{K}_K(\mathbf{X}^T\mathbf{X}, \mathbf{X}^T\mathbf{y}),$$

and by Lemma 1 an equivalent reformulation is thus

$$\underset{\boldsymbol{\alpha}\in\mathbb{R}^K}{\text{minimise}} \quad \frac{1}{2n}\|\mathbf{y} - \mathbf{X}\mathbf{K}\boldsymbol{\alpha}\|_2^2, \tag{3}$$

where $\mathbf{K} \in \mathbb{R}^{p\times K}$ again is a basis for the Krylov subspace $\mathcal{K}_K(\mathbf{X}^T\mathbf{X}, \mathbf{X}^T\mathbf{y})$ and $\boldsymbol{\alpha} \in \mathbb{R}^K$. We assume that $\mathbf{K}$ is an orthonormal basis. An analytical solution is thus

$$\boldsymbol{\alpha} = (\mathbf{K}^T\mathbf{X}^T\mathbf{X}\mathbf{K})^{-1}\mathbf{K}^T\mathbf{X}^T\mathbf{y},$$

assuming that $\mathbf{K}^T\mathbf{X}^T\mathbf{X}\mathbf{K}$ is invertible, but any numerical optimisation algorithm that solves Equation 3 can of course also be used. Finally, the PLS-R regression coefficient vector is retrieved as

$$\boldsymbol{\beta}_{\text{PLS}} = \mathbf{K}\boldsymbol{\alpha}.$$

Note that this is the *same* regression vector as that found in Equation 2.

## 2.3 Regularising the Regression Vector in Partial Least Squares Regression

With Equation 3, the PLS-R problem is in a familiar form, and we can apply any regularisation we want to the least squares objective. *E.g.*, we can add a square $\ell_2$ norm penalty, and obtain a ridge PLS-R hybrid model, where the $\ell_2$ regularisation parameter and $K$ would control the trade-off between linear least squares regression, ridge regression, and PLS-R. Equivalently, we can add an $\ell_1$ norm penalty, for a Lasso PLS-R hybrid model, with a trade-off between linear least squares regression, the Lasso, and PLS-R. This is particularly interesting, since the $\ell_1$ norm penalty performs variable selection, and thus a truly sparse PLS-R model.

We chose to add both the $\ell_1$ and squared $\ell_2$ norm penalties, to obtain an elastic net (Zou & Hastie, 2005) PLS-R hybrid model, where we thus can find an optimal trade-off between $\ell_1$, squared $\ell_2$, and PLS-R regularisation of the regression coefficient vector, *i.e.* the $\mathbf{K}\boldsymbol{\alpha}$. This model thus also performs variable selection in the regression coefficient vector, *i.e.* a sparse PLS-R model, where the sparsity is with respect to the regression coefficients instead of the weights and/or scores. The optimisation problem thus becomes

$$\underset{\boldsymbol{\alpha}\in\mathbb{R}^K}{\text{minimise}} \frac{1}{2n}\|\mathbf{y} - \mathbf{X}\mathbf{K}\boldsymbol{\alpha}\|_2^2 + \frac{\gamma}{2}\|\mathbf{K}\boldsymbol{\alpha}\|_2^2 + \lambda\|\mathbf{K}\boldsymbol{\alpha}\|_1, \tag{4}$$

where $\lambda > 0$ and $\gamma > 0$ are regularisation parameters (or rather, Lagrange multipliers) controlling the trade-off between the main objective and the regularisation terms. We obtain our final sparse regression vector as $\widehat{\boldsymbol{\beta}}_{\text{PLS}} = \mathbf{K}\boldsymbol{\alpha}$.

The optimisation problem in Equation 4 was the main object of our attention in this work, and in the examples that follow, we used the alternating direction method of multipliers (ADMM, Gabay & Mercier, 1976; Boyd et al., 2010) to solve it.

### 2.3.1 The Steps of the Alternating Direction Method of Multipliers

We cast the program in Equation 4 in the following form,

$$\underset{\boldsymbol{\alpha}\in\mathbb{R}^K}{\text{minimise}} \quad \frac{1}{2n}\|\mathbf{y}-\mathbf{X}\mathbf{K}\boldsymbol{\alpha}\|_2^2 + \frac{\gamma}{2}\|\mathbf{K}\boldsymbol{\alpha}\|_2^2 + \lambda\|\mathbf{z}\|_1 \tag{5}$$

$$\text{subject to} \quad \mathbf{K}\mathbf{x} = \mathbf{z}.$$

We formulate the augmented Lagrangian of Equation 5,

$$L_\rho(\mathbf{x},\mathbf{z},\boldsymbol{v}) = \frac{1}{2n}\|\mathbf{y}-\mathbf{X}\mathbf{K}\mathbf{x}\|_2^2 + \frac{\gamma}{2}\|\mathbf{K}\mathbf{x}\|_2^2 + \lambda\|\mathbf{z}\|_1 + \boldsymbol{v}^{\mathrm{T}}(\mathbf{K}\mathbf{x}-\mathbf{z}) + \frac{\rho}{2}\|\mathbf{K}\mathbf{x}-\mathbf{z}\|_2^2, \tag{6}$$

with $\rho > 0$ the penalty parameter and $\boldsymbol{v} \in \mathbb{R}^p$ a vector of Lagrange multipliers. For the ADMM algorithm, we must minimise Equation 6 with respect to $\mathbf{x}$ and with respect to $\mathbf{z}$. We see that, by setting the gradient of $L_\rho$ with respect to $\mathbf{x}$ to zero and solving for $\mathbf{x}$, we obtain

$$\underset{\mathbf{x}\in\mathbb{R}^K}{\arg\min} \; L_\rho(\mathbf{x},\mathbf{z},\boldsymbol{v}) = \left(\frac{1}{n}\mathbf{K}^{\mathrm{T}}\mathbf{X}^{\mathrm{T}}\mathbf{X}\mathbf{K} + (\gamma+\rho)\mathbf{K}^{\mathrm{T}}\mathbf{K}\right)^{-1}\left(\frac{1}{n}\mathbf{K}^{\mathrm{T}}\mathbf{X}^{\mathrm{T}}\mathbf{y} - \mathbf{K}^{\mathrm{T}}\boldsymbol{v} + \rho\mathbf{K}^{\mathrm{T}}\mathbf{z}\right).$$

To minimise $L_\rho$ with respect to $\mathbf{z}$, we first see that

$$L_\rho(\mathbf{x},\mathbf{z},\boldsymbol{v}) = \lambda\|\mathbf{z}\|_1 - \boldsymbol{v}^{\mathrm{T}}\mathbf{z} + \frac{\rho}{2}\|\mathbf{K}\mathbf{x}-\mathbf{z}\|_2^2 + C_1(\mathbf{x},\boldsymbol{v})$$

$$= \lambda\|\mathbf{z}\|_1 + \frac{\rho}{2}\left\|\left(\mathbf{K}\mathbf{x}+\frac{\boldsymbol{v}}{\rho}\right)-\mathbf{z}\right\|_2^2 + C_2(\mathbf{x},\boldsymbol{v})$$

where $C_1$ and $C_2$ are constant *wrt.* $\mathbf{z}$. We recognise this as the proximal operator of the $\ell_1$ norm, and thus arrive at

$$\underset{\mathbf{z}\in\mathbb{R}^p}{\arg\min} \; L_\rho(\mathbf{x},\mathbf{z},\boldsymbol{v}) = \text{prox}_{\frac{\lambda}{\rho}\|\cdot\|_1}\left(\mathbf{K}\mathbf{x}+\frac{\boldsymbol{v}}{\rho}\right).$$

Finally, the steps of ADMM are,

$$\mathbf{x}^{(s+1)} = \underset{\mathbf{x}\in\mathbb{R}^K}{\arg\min} \; L_\rho(\mathbf{x},\mathbf{z}^{(s)},\boldsymbol{v}^{(s)})$$

$$\mathbf{z}^{(s+1)} = \underset{\mathbf{z}\in\mathbb{R}^p}{\arg\min} \; L_\rho(\mathbf{x}^{(s+1)},\mathbf{z},\boldsymbol{v}^{(s)})$$

$$\boldsymbol{v}^{(s+1)} = \boldsymbol{v}^{(s)} + \rho(\mathbf{K}\mathbf{x}^{(s+1)}-\mathbf{z}^{(s+1)}).$$

## 2.4 Reconstructing the Components of Partial Least Squares Regression

The regression coefficient vector we obtain by solving Equation 4 will not coincide with the one obtained by PLS-R in Equation 2, nor with the equivalent one we obtain by solving Equation 3. While we still select it from the Krylov subspace $\mathcal{K}_K(\mathbf{X}^{\mathrm{T}}\mathbf{X}, \mathbf{X}^{\mathrm{T}}\mathbf{y})$, the found vector, $\boldsymbol{\alpha}$, will be different in Equation 4 from that found in Equation 3, and so the corresponding regression vectors will also be different.

What we can do is to set up an optimisation problem, searching for weights, scores, and loadings fulfilling the properties of the components of PLS-R, at least approximately, and that gives a regression vector that

is *close* to $\widehat{\boldsymbol{\beta}}_{\mathrm{PLS}}$. We therefore want to solve the following optimisation problem,

$$\underset{\widetilde{\mathbf{w}}_{k+1}\in\mathbb{R}^p}{\text{maximise}} \quad \mathbf{y}^{\mathrm{T}}\mathbf{X}\widetilde{\mathbf{w}}_{k+1} \tag{7}$$

$$\text{subject to} \quad \|\widetilde{\mathbf{w}}_{k+1}\|_2^2 = 1, \tag{8}$$

$$\widetilde{\mathbf{W}}_k^{\mathrm{T}}\widetilde{\mathbf{w}}_{k+1} = \mathbf{0}, \tag{9}$$

$$\widetilde{\mathbf{T}}_k^{\mathrm{T}}\widetilde{\mathbf{t}}_{k+1} = \mathbf{0}, \tag{10}$$

$$\widetilde{\mathbf{W}}_k^{\mathrm{T}}\widetilde{\mathbf{p}}_{k+1} = \mathbf{0}, \tag{11}$$

$$\|\mathbf{w}_{k+1} - \widetilde{\mathbf{w}}_{k+1}\|_2^2 \le \alpha, \tag{12}$$

$$\|\mathbf{t}_{k+1} - \widetilde{\mathbf{t}}_{k+1}\|_2^2 \le \beta, \tag{13}$$

$$\|\mathbf{p}_{k+1} - \widetilde{\mathbf{p}}_{k+1}\|_2^2 \le \gamma, \tag{14}$$

$$\|c_{k+1} - \widetilde{c}_{k+1}\|_2^2 \le \delta, \tag{15}$$

$$\|\widehat{\boldsymbol{\beta}}_{\mathrm{PLS}} - \widetilde{\mathbf{W}}_{k+1}(\widetilde{\mathbf{P}}_{k+1}^{\mathrm{T}}\widetilde{\mathbf{W}}_{k+1})^{-1}\widetilde{\mathbf{C}}_{k+1}^{\mathrm{T}}\|_2^2 \le \varepsilon, \tag{16}$$

where parameters with a tilde, such as $\widetilde{\mathbf{w}}_{k+1}$, are the component $k+1$ parameters that we want to find. Parameters without tilde, such as $\mathbf{w}$, are those found using unregularised PLS-R (from Section 2.1), and $\widehat{\boldsymbol{\beta}}_{\mathrm{PLS}} = \mathbf{K}\boldsymbol{\alpha}$ is the regularised PLS-R regression vector found by Equation 5. A matrix with an index has a number of columns containing the components already found in order, *e.g.* $\widetilde{\mathbf{W}}_k = [\widetilde{\mathbf{w}}_1, \dots, \widetilde{\mathbf{w}}_k]$. Hence, $\widetilde{\mathbf{W}}_{k+1}$ also includes the current sought parameter vector.

The objective, Equation 7, is the original PLS-R objective (from Equation 1), which still is what we want to maximise. Equation 8 is the unit norm constraint. Equation 9 is an orthogonality constraint on the weight vector, $\widetilde{\mathbf{w}}_{k+1}$, to the $k$ already found weight vectors, $\widetilde{\mathbf{W}}_k$. Equation 10 is a corresponding orthogonality constraint for the score vectors. The loadings, $\widetilde{\mathbf{p}}_{k+1}$, should be orthogonal to the already found weight vectors, encoded in Equation 11 (see *e.g.*, Manne, 1987). Further, we may want the found weights, scores, and loadings to be close to the PLS-R weights, scores, and loadings, which is encoded in Equations 12–15. Finally, Equation 16 forces the regression coefficient vector, computed using Equation 2, to be near to the one obtained from the regularised Krylov formulation of the PLS-R problem in Equation 5.

This formulation, in Equation 7, poses the exact problem that we want to solve, but it becomes a very difficult problem in practice. We have multiple constraints of which several are non-linear, non-convex, and there may not even be a non-empty feasible set. We therefore propose the following slightly relaxed problem, employing the method of Lagrange multipliers,

$$\underset{\widetilde{\boldsymbol{\omega}}_{k+1}\in\mathbb{R}^p}{\text{minimise}} \quad f(\widetilde{\boldsymbol{\omega}}_{k+1}) \tag{17}$$

$$= -\mathbf{y}^{\mathrm{T}}\mathbf{X}\mathbf{K}\widetilde{\boldsymbol{\omega}}_{k+1} \tag{18}$$

$$+ \lambda\|\widehat{\boldsymbol{\beta}}_{\mathrm{PLS}} - \widetilde{\mathbf{W}}_{k+1}(\widetilde{\mathbf{P}}_{k+1}^{\mathrm{T}}\widetilde{\mathbf{W}}_{k+1})^{-1}\widetilde{\mathbf{C}}_{k+1}^{\mathrm{T}}\|_2^2 \tag{19}$$

$$+ \mu\|\mathbf{w}_{k+1} - \mathbf{K}\widetilde{\boldsymbol{\omega}}_{k+1}\|_2^2 \tag{20}$$

$$+ \nu\|\mathbf{t}_{k+1} - \mathbf{X}\mathbf{K}\widetilde{\boldsymbol{\omega}}_{k+1}\|_2^2 \tag{21}$$

$$+ \xi\|(\mathbf{t}_{k+1}^{\mathrm{T}}\mathbf{t}_{k+1})\mathbf{p}_{k+1} - \mathbf{X}^{\mathrm{T}}\mathbf{X}\mathbf{K}\widetilde{\boldsymbol{\omega}}_{k+1}\|_2^2 \tag{22}$$

$$+ \pi\|(\mathbf{t}_{k+1}^{\mathrm{T}}\mathbf{t}_{k+1})c_{k+1} - \mathbf{y}^{\mathrm{T}}\mathbf{X}\mathbf{K}\widetilde{\boldsymbol{\omega}}_{k+1}\|_2^2 \tag{23}$$

$$\text{subject to} \quad \|\mathbf{K}\widetilde{\boldsymbol{\omega}}_{k+1}\|_2^2 \le 1, \tag{24}$$

$$\widetilde{\mathbf{W}}_k^{\mathrm{T}}\mathbf{K}\widetilde{\boldsymbol{\omega}}_{k+1} = \mathbf{0}, \tag{25}$$

$$\widetilde{\mathbf{T}}_k^{\mathrm{T}}\mathbf{X}\mathbf{K}\widetilde{\boldsymbol{\omega}}_{k+1} = \mathbf{0}, \tag{26}$$

$$\widetilde{\mathbf{W}}_k^{\mathrm{T}}\mathbf{X}^{\mathrm{T}}\mathbf{X}\mathbf{K}\widetilde{\boldsymbol{\omega}}_{k+1} = \mathbf{0}, \tag{27}$$

where $\widetilde{\mathbf{w}}_{k+1} = \mathbf{K}\widetilde{\boldsymbol{\omega}}_{k+1}$, the $\lambda, \mu, \nu, \xi$, and $\pi$ are regularisation parameters (Lagrange multipliers), and where the constraints from Equations 8–16, for which we the projection operators are easy to compute, have been

kept as constraints in Equations 24–27, and those that are more difficult, or alternatively, very easy to optimise over in their penalty form, have been put as penalty terms instead.

We now give the derivation and interpretation of the terms in Equations 19–27 in order:

- Equation 19: Note that $\widetilde{\mathbf{W}}_{k+1} = [\widetilde{\mathbf{w}}_1, \ldots, \widetilde{\mathbf{w}}_k, \mathbf{K}\widetilde{\boldsymbol{\omega}}_{k+1}]$, and similar for $\widetilde{\mathbf{P}}_{k+1}$ and $\widetilde{\mathbf{C}}_{k+1}$. Corresponding to Equation 16, this is a non-linear function in $\widetilde{\boldsymbol{\omega}}_{k+1}$, for which we don't know the projection operator nor the proximal operator. It is easier to minimise in penalty form.

- Equation 20: Corresponds to Equation 12. This projection operator is known and easy to compute, and a smooth function in $\widetilde{\boldsymbol{\omega}}_{k+1}$, why we make it into a penalty instead.

- Equation 21: Corresponds to Equation 13. We express this as a function of $\widetilde{\boldsymbol{\omega}}_{k+1}$, with $\widetilde{\mathbf{t}}_{k+1} = \mathbf{X}\mathbf{K}\widetilde{\boldsymbol{\omega}}_{k+1}$, and since it is smooth and convex, we put it as a penalty.

- Equation 22: Corresponds to Equation 14. Note that PLS-R defines the loadings as

$$\widetilde{\mathbf{p}}_{k+1} = \frac{\mathbf{X}^{\mathrm{T}}\widetilde{\mathbf{t}}_{k+1}}{\widetilde{\mathbf{t}}_{k+1}^{\mathrm{T}}\widetilde{\mathbf{t}}_{k+1}} = \frac{\mathbf{X}^{\mathrm{T}}\mathbf{X}\widetilde{\mathbf{w}}_{k+1}}{\|\mathbf{X}\widetilde{\mathbf{w}}_{k+1}\|_2^2} = \frac{\mathbf{X}^{\mathrm{T}}\mathbf{X}\mathbf{K}\widetilde{\boldsymbol{\omega}}_{k+1}}{\|\mathbf{X}\mathbf{K}\widetilde{\boldsymbol{\omega}}_{k+1}\|_2^2},$$

which is non-linear in $\widetilde{\boldsymbol{\omega}}_{k+1}$. To get rid of the denominator we make an approximation in that we multiply both terms by the squared norm of their corresponding score vector, and instead ask that $\mathbf{X}^{\mathrm{T}}\mathbf{X}\mathbf{K}\widetilde{\boldsymbol{\omega}}_{k+1}$ be close to $(\mathbf{t}_{k+1}^{\mathrm{T}}\mathbf{t}_{k+1})\mathbf{p}_{k+1}$. Hence, another smooth convex function put as a penalty.

- Equation 23: Corresponds to Equation 15. Recall that PLS-R defines the $y$-loadings as

$$\widetilde{c}_{k+1} = \frac{\mathbf{y}^{\mathrm{T}}\widetilde{\mathbf{t}}_{k+1}}{\widetilde{\mathbf{t}}_{k+1}^{\mathrm{T}}\widetilde{\mathbf{t}}_{k+1}} = \frac{\mathbf{y}^{\mathrm{T}}\mathbf{X}\widetilde{\mathbf{w}}_{k+1}}{\|\mathbf{X}\widetilde{\mathbf{w}}_{k+1}\|_2^2} = \frac{\mathbf{y}^{\mathrm{T}}\mathbf{X}\mathbf{K}\widetilde{\boldsymbol{\omega}}_{k+1}}{\|\mathbf{X}\mathbf{K}\widetilde{\boldsymbol{\omega}}_{k+1}\|_2^2},$$

which is non-linear in $\widetilde{\boldsymbol{\omega}}_{k+1}$. To get rid of the denominator, we again make an approximation in that we multiply both terms by the squared norm of their corresponding score vector, and ask that $\mathbf{y}^{\mathrm{T}}\mathbf{X}\mathbf{K}\widetilde{\boldsymbol{\omega}}_{k+1}$ be close to $(\mathbf{t}_{k+1}^{\mathrm{T}}\mathbf{t}_{k+1})c_{k+1}$. Hence, a smooth convex function in $\widetilde{\boldsymbol{\omega}}_{k+1}$, put as a penalty.

- Equation 24: This is a convex relaxation of Equation 8.

- Equations 25–26: Same as Equations 9–10, with the difference that Equation 26 is expressed as a function of $\widetilde{\boldsymbol{\omega}}_{k+1}$.

- Equation 27: Corresponds to Equation 11. We know that the PLS-R loadings satisfy (Höskuldsson, 2003),

$$\widetilde{\mathbf{W}}_k^{\mathrm{T}}\widetilde{\mathbf{p}}_{k+1} = \frac{\widetilde{\mathbf{W}}_k^{\mathrm{T}}\mathbf{X}^{\mathrm{T}}\widetilde{\mathbf{t}}_{k+1}}{\widetilde{\mathbf{t}}_{k+1}^{\mathrm{T}}\widetilde{\mathbf{t}}_{k+1}} = \frac{\widetilde{\mathbf{W}}_k^{\mathrm{T}}\mathbf{X}^{\mathrm{T}}\mathbf{X}\mathbf{K}\widetilde{\boldsymbol{\omega}}_{k+1}}{\|\mathbf{X}\mathbf{K}\widetilde{\boldsymbol{\omega}}_{k+1}\|_2^2} = \mathbf{0},$$

which clearly is equivalent to

$$\widetilde{\mathbf{W}}_k^{\mathrm{T}}\widetilde{\mathbf{p}}_{k+1} = \widetilde{\mathbf{W}}_k^{\mathrm{T}}\mathbf{X}^{\mathrm{T}}\mathbf{X}\mathbf{K}\widetilde{\boldsymbol{\omega}}_{k+1} = \mathbf{0},$$

assuming $\widetilde{\boldsymbol{\omega}}_{k+1}^{\mathrm{T}}\mathbf{K}^{\mathrm{T}}\mathbf{X}^{\mathrm{T}}\mathbf{X}\mathbf{K}\widetilde{\boldsymbol{\omega}}_{k+1} > 0$, but since $K \leq \mathrm{rank}(\mathbf{X})$, this is achieved.

With these changes, consisting of several reformulations, one convex relaxation, and two approximations, we have an objective function that is the sum of a number of functions that all but one are convex, with four convex constraints (of which three are linear). We can solve this problem using *e.g.* projected gradient descent (Bertsekas, 1999), or any other optimisation algorithm of choice.

In order to apply projected gradient descent, we need to know the gradient of $f$ and the projection operator corresponding to the four constraints in Equations 24–27. These are straight-forward, but need to be outlined in more detail.

### 2.4.1 Projection Operators

Each constraint in Equations 24–27 correspond to a set of feasible points, denoted $\mathcal{S}_1, \ldots, \mathcal{S}_4$, respectively. In order for all four constraints to be satisfied, the solution must lie in all of them, *i.e.* we seek a point that lie in their intersection,

$$\mathcal{S} = \{\mathbf{x} : \mathbf{x} \in \mathcal{S}_1 \wedge \cdots \wedge \mathbf{x} \in \mathcal{S}_4\} = \left\{\mathbf{x} : \mathbf{x} \in \bigcap_{i=1}^{4} \mathcal{S}_i\right\}.$$

We see from Equations 24–27 that for all $\mathcal{S}_i$, for $i = 1, \ldots, 4$, we have that at least $\mathbf{0} \in \mathcal{S}_i$, and so $\mathcal{S} \neq \emptyset$ We note that each $\mathcal{S}_i$, for $i = 1, \ldots, 4$, is a convex set, and since the intersection of convex sets is convex, $\mathcal{S}$ is also a convex set.

The single projection operator corresponding to the four constraints in Equations 24–27 is the projection onto their intersection, *i.e.* the projection onto $\mathcal{S}$. The projection of a point $\mathbf{w} \in \mathbb{R}^p$ onto a convex set, $\mathcal{S} \subseteq \mathbb{R}^p$, is defined as,

$$\operatorname{proj}_{\mathcal{S}}(\mathbf{w}) = \underset{\mathbf{x} \in \mathbb{R}^p}{\arg\min} \|\mathbf{x} - \mathbf{w}\|_2^2 + \chi_{\mathcal{S}}(\mathbf{x}), \tag{28}$$

where $\chi_{\mathcal{S}}$ is the characteristic function over $\mathcal{S}$, *i.e.*,

$$\chi_{\mathcal{S}}(\mathbf{x}) = \begin{cases} 0 & \text{if } \mathbf{x} \in \mathcal{S}, \\ \infty & \text{if } \mathbf{x} \notin \mathcal{S}. \end{cases}$$

We can numerically compute the projection onto the intersection of the four sets, $\mathcal{S}_1, \ldots \mathcal{S}_4$, *i.e.* onto $\mathcal{S}$, by using a parallel Dykstra-like proximal algorithm, as outlined by *e.g.* Combettes & Pesquet (2011).

We give the two projection operators, and start with Equation 24. The proximal operator for $\lambda\|\mathbf{K}\mathbf{x}\|_2^2$ is trivially

$$\operatorname{prox}_{\lambda\|\mathbf{K}\cdot\|_2^2}(\mathbf{x}) = (\mathbf{I} + 2\lambda\mathbf{K}^{\mathrm{T}}\mathbf{K})^{-1}\mathbf{x} = \frac{1}{1 + 2\lambda}\mathbf{x},$$

since $\mathbf{K}$ is assumed orthonormal (and thus $\mathbf{K}^{\mathrm{T}}\mathbf{K} = \mathbf{I}$), and we seek the smallest $\lambda$ such that Equation 24 is fulfilled, *i.e.*, such that $\mathbf{x} \in \mathcal{S}_1 = \{\mathbf{x} \in \mathbb{R}^p : \|\mathbf{K}\operatorname{prox}_{\lambda\|\mathbf{K}\cdot\|_2^2}(\mathbf{x})\|_2^2 \leq 1\}$, which we achieve by finding the smallest $\lambda^*$ such that

$$\left\|\mathbf{K}\left(\frac{1}{1 + 2\lambda^*}\mathbf{x}\right)\right\|_2^2 \leq 1 \quad \Longleftrightarrow \quad \lambda^* \geq \frac{\|\mathbf{K}\mathbf{x}\|_2}{2} - \frac{1}{2}.$$

Hence, using this $\lambda^*$, the projection operator becomes

$$\operatorname{proj}_{\mathcal{S}_1}(\mathbf{x}) = \begin{cases} \mathbf{x} & \text{if } \|\mathbf{K}\mathbf{x}\|_2^2 \leq 1, \\ \frac{1}{1 + 2\lambda^*}\mathbf{x} & \text{otherwise.} \end{cases}$$

The constraints in Equations 25–27 all have the general form $\mathbf{A}_i\mathbf{x} = \mathbf{0}$ with $\mathbf{A}_2 = \widetilde{\mathbf{W}}_k^{\mathrm{T}}\mathbf{K}$, $\mathbf{A}_3 = \widetilde{\mathbf{T}}_k^{\mathrm{T}}\mathbf{X}\mathbf{K}$, or $\mathbf{A}_4 = \widetilde{\mathbf{W}}_k^{\mathrm{T}}\mathbf{X}^{\mathrm{T}}\mathbf{X}\mathbf{K}$, respectively. The projection operator onto the sets $\mathcal{S}_i = \{\mathbf{x} \in \mathbb{R}^p : \mathbf{A}_i\mathbf{x} = \mathbf{0}\}$, for $i \in \{2, 3, 4\}$, has the analytic solution (Bauschke & Kruk, 2004),

$$\operatorname{proj}_{\mathcal{S}_i}(\mathbf{x}) = \begin{cases} \mathbf{x}, & \text{if } \mathbf{A}_i\mathbf{x} = \mathbf{0}, \\ \mathbf{x} - \mathbf{A}_i^{\dagger}\mathbf{A}_i\mathbf{x}, & \text{otherwise,} \end{cases}$$

where $\mathbf{A}^{\dagger}$ is the Moore-Penrose pseudo-inverse of $\mathbf{A}$.

### 2.4.2 Gradient of the Objective

We need to compute the gradient of $f$ in Equation 17. Now, Equation 18 and Equations 20–23 are straight-forward linear and quadratic functions, with trivial gradients, but Equation 19 is more difficult to find, why we present it here.

We have that (Höskuldsson, 2003),

$$
\begin{aligned}
\widetilde{\mathbf{w}}_{k+1}^* &= \widetilde{\mathbf{w}}_{k+1} - \left[ \sum_{i=1}^{k} \widetilde{\mathbf{w}}_i^* \widetilde{\mathbf{p}}_i^{\mathrm{T}} \right] \widetilde{\mathbf{w}}_{k+1} \\
&= \mathbf{K} \widetilde{\boldsymbol{\omega}}_{k+1} - \left[ \sum_{i=1}^{k} \widetilde{\mathbf{w}}_i^* \widetilde{\mathbf{p}}_i^{\mathrm{T}} \right] \mathbf{K} \widetilde{\boldsymbol{\omega}}_{k+1} \\
&= (\mathbf{I} - \mathbf{D}) \mathbf{K} \widetilde{\boldsymbol{\omega}}_{k+1},
\end{aligned}
$$

where $\mathbf{I}$ is an identity matrix and $\mathbf{D}$ is a constant matrix.

Let $\widehat{\boldsymbol{\beta}}_{\mathrm{PLS},k}$ be the regression coefficient vector approximated using the $k$ previously found components, and further let $\mathbf{a} = \mathbf{K}^{\mathrm{T}} \mathbf{X}^{\mathrm{T}} \mathbf{y}$, $\mathbf{b} = \widehat{\boldsymbol{\beta}}_{\mathrm{PLS}} - \widehat{\boldsymbol{\beta}}_{\mathrm{PLS},k}$, $\mathbf{A} = (\mathbf{I} - \mathbf{D}) \mathbf{K}$, and $\mathbf{B} = \mathbf{K}^{\mathrm{T}} \mathbf{X}^{\mathrm{T}} \mathbf{X} \mathbf{K}$. Then the gradient becomes,

$$
\begin{aligned}
\nabla_{\widetilde{\boldsymbol{\omega}}_{k+1}} & \left\| \widehat{\boldsymbol{\beta}}_{\mathrm{PLS}} - \widetilde{\mathbf{W}}_{k+1} (\widetilde{\mathbf{P}}_{k+1}^{\mathrm{T}} \widetilde{\mathbf{W}}_{k+1})^{-1} \widetilde{\mathbf{C}}_{k+1}^{\mathrm{T}} \right\|_2^2 \quad (29) \\
&= \frac{-2(\mathbf{A}^{\mathrm{T}} \mathbf{b} \mathbf{a}^{\mathrm{T}} + \mathbf{a} \mathbf{b}^{\mathrm{T}} \mathbf{A}) \widetilde{\boldsymbol{\omega}}_{k+1}}{\widetilde{\boldsymbol{\omega}}_{k+1}^{\mathrm{T}} \mathbf{B} \widetilde{\boldsymbol{\omega}}_{k+1}} \\
&\quad + \frac{4 \mathbf{b}^{\mathrm{T}} \mathbf{A} \widetilde{\boldsymbol{\omega}}_{k+1} \widetilde{\boldsymbol{\omega}}_{k+1}^{\mathrm{T}} \mathbf{a} \mathbf{B} \widetilde{\boldsymbol{\omega}}_{k+1}}{(\widetilde{\boldsymbol{\omega}}_{k+1}^{\mathrm{T}} \mathbf{B} \widetilde{\boldsymbol{\omega}}_{k+1})^2} \\
&\quad + \frac{2(\mathbf{A}^{\mathrm{T}} \mathbf{A} \widetilde{\boldsymbol{\omega}}_{k+1} \widetilde{\boldsymbol{\omega}}_{k+1}^{\mathrm{T}} \mathbf{a} + \widetilde{\boldsymbol{\omega}}_{k+1}^{\mathrm{T}} \mathbf{A}^{\mathrm{T}} \mathbf{A} \widetilde{\boldsymbol{\omega}}_{k+1} \mathbf{a}) \mathbf{a}^{\mathrm{T}} \widetilde{\boldsymbol{\omega}}_{k+1}}{(\widetilde{\boldsymbol{\omega}}_{k+1}^{\mathrm{T}} \mathbf{B} \widetilde{\boldsymbol{\omega}}_{k+1})^2} \\
&\quad - \frac{4 \mathbf{a}^{\mathrm{T}} \widetilde{\boldsymbol{\omega}}_{k+1} \widetilde{\boldsymbol{\omega}}_{k+1}^{\mathrm{T}} \mathbf{A}^{\mathrm{T}} \mathbf{A} \widetilde{\boldsymbol{\omega}}_{k+1} \widetilde{\boldsymbol{\omega}}_{k+1}^{\mathrm{T}} \mathbf{a} \mathbf{B} \widetilde{\boldsymbol{\omega}}_{k+1}}{(\widetilde{\boldsymbol{\omega}}_{k+1}^{\mathrm{T}} \mathbf{B} \widetilde{\boldsymbol{\omega}}_{k+1})^3}.
\end{aligned}
$$

### 2.4.3 Minimising the Objective Function

We used projected gradient descent (Bertsekas, 1999) to solve the non-linear program in Equation 17, which amounts to iterating the weight update scheme,

$$
\widetilde{\boldsymbol{\omega}}_{k+1}^{(s+1)} \leftarrow \mathrm{proj}_{\mathcal{S}} \big( \widetilde{\boldsymbol{\omega}}_{k+1}^{(s)} - \eta \nabla f(\widetilde{\boldsymbol{\omega}}_{k+1}^{(s)}) \big)
$$

where $s$ is a sequence index, $\eta > 0$ is a step size, the gradient of $f$ is Equation 29 plus the gradients of the rest of the terms, *i.e.* from Equation 18 and Equations 20–23, and the projection operator, $\mathrm{proj}_{\mathcal{S}}$, was computed numerically by solving Equation 28 using a parallel Dykstra-like proximal algorithm (Combettes & Pesquet, 2011).

### 2.4.4 The Found Regression Coefficient Vector

We have the following immediate result about the regression coefficient vector, $\widehat{\boldsymbol{\beta}}_{\mathrm{PLS},K} = \widetilde{\mathbf{W}}_K (\widetilde{\mathbf{P}}_K^{\mathrm{T}} \widetilde{\mathbf{W}}_K)^{-1} \widetilde{\mathbf{C}}_K^{\mathrm{T}}$, found by using the program in Equation 17.

**Theorem 1.** *The PLS-R regression vector, $\widehat{\boldsymbol{\beta}}_{\mathrm{PLS},K}$, found through Equation 17, lie in the Krylov subspace of order $K$ generated by $\mathbf{X}^{\mathrm{T}} \mathbf{X}$ and $\mathbf{X}^{\mathrm{T}} \mathbf{y}$, i.e.*

$$
\widehat{\boldsymbol{\beta}}_{\mathrm{PLS},K} \in \mathcal{K}_K(\mathbf{X}^{\mathrm{T}} \mathbf{X}, \mathbf{X}^{\mathrm{T}} \mathbf{y}).
$$

*Proof.* All weight vectors found through Equation 17 are written in the form $\widetilde{\mathbf{w}}_k = \mathbf{K} \widetilde{\boldsymbol{\omega}}_k$, where $\mathbf{K}$ is a basis for the Krylov subspace $\mathcal{K}_K(\mathbf{X}^{\mathrm{T}} \mathbf{X}, \mathbf{X}^{\mathrm{T}} \mathbf{y})$. We can then write

$$
\widetilde{\mathbf{W}}_K = \mathbf{K} \boldsymbol{\Omega}_K,
$$

with $\boldsymbol{\Omega}_K = [\widetilde{\boldsymbol{\omega}}_1, \ldots, \widetilde{\boldsymbol{\omega}}_K]$. *I.e.*, all weight vectors lie in the Krylov subspace. Hence, by Lemma 1, the PLS-R regression vector, $\widehat{\boldsymbol{\beta}}_{\mathrm{PLS},K}$, lie in that Krylov subspace of order $K$ generated by $\mathbf{X}^{\mathrm{T}} \mathbf{X}$ and $\mathbf{X}^{\mathrm{T}} \mathbf{y}$. $\qquad \square$

Hence, the solution found using the Krylov formulation has the same property as the PLS-R solution, namely that both the weights, $\widetilde{\mathbf{W}}$, and the regression vector, $\widehat{\boldsymbol{\beta}}_{\text{PLS}}$, lie in a Krylov subspace. This property may lead to better future algorithms for solving the PLS-R problems.

## 3 Examples

To illustrate the utility of the proposed Krylov-based PLS-R formulation, we present two examples. The first example is based on simulated data and the second example is based on near infrared reflectance (NIR) scans of soil samples. Both examples are analysed *without* regularisation, to show that the proposed method is able to reconstruct the scores and loadings of a standard PLS-R model computed using the NIPALS algorithm (Helland, 1988; Abdi, 2010), and both are also analysed *with* elastic-net ($\ell_1$ and squared $\ell_2$) regularisation, to illustrate the extended regularisation, the variable selection, and the interpretation of the regression coefficient vector and the reconstructed scores and loadings.

### 3.1 Example 1: Simulated Data

The first example illustrates how the proposed regularised PLS-R method works in comparison to standard PLS-R. We will illustrate that without regularisation, the proposed method gives the same result as standard PLS-R, and further how the regularised PLS-R differs from the regular PLS-R in terms of the regression vector, and the weights and scores.

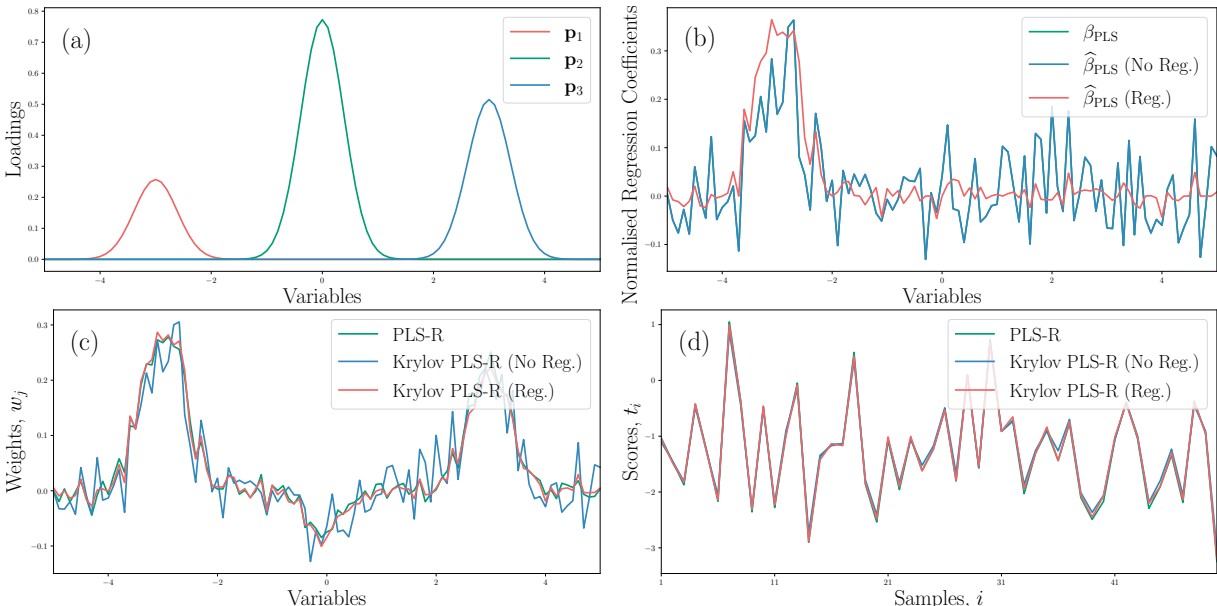

Figure 1: (a) The loading profiles that make up the data matrix. (b) The regression coefficient vectors found using regular PLS-R. Note that the green and blue curves are indistinguishable. (c) The first weight vectors for each method. (d) The first score vectors for each method.

The data were collected in a matrix, $\mathbf{X} \in \mathbb{R}^{n \times p}$, with $n = 50$ and $p = 101$, and was composed of three spectra, each with a Gaussian profile as seen in Figure 1 (a). The data were constructed as

$$\mathbf{X} = \mathbf{t}_1 \mathbf{p}_1^{\text{T}} + \mathbf{t}_2 \mathbf{p}_2^{\text{T}} + \mathbf{t}_3 \mathbf{p}_3^{\text{T}} + \mathbf{E},$$

where $\mathbf{y} = \mathbf{t}_1 \sim \mathcal{N}(\mathbf{0}_n, \mathbf{I}_n)$ were sampled from a standard normal, $\mathbf{t}_2 = \frac{1}{2}|\mathbf{z}_2| + 0.35$ with $\mathbf{z}_2 \sim \mathcal{N}(\mathbf{0}_n, \mathbf{I}_n)$, and $\mathbf{t}_3 = \mathbf{z}_3 - 1.25$ with $\mathbf{z}_3 \sim \mathcal{N}(\mathbf{0}_n, \mathbf{I}_n)$, and $\mathbf{E}$ are independent zero-mean normal with variance 0.01. There was thus a perfect correlation between $\mathbf{y}$ and $\mathbf{t}_1$, the correlation between $\mathbf{y}$ and $\mathbf{t}_2$ was about 0.213, and the correlation between $\mathbf{y}$ and $\mathbf{t}_3$ was about 0.005.

We fit one regular PLS-R model, one PLS-R model based on the Krylov subspace formulation without regularisation, and one PLS-R model based on the Krylov subspace formulation with elastic net regularisation. The number of components extracted were $K = 20$, and for the elastic net regularisation, we used $\gamma = 0.00125$ and $\lambda = 0.02375$.

The PLS-R regression vectors are illustrated in Figure 1 (b). We see that the regular PLS-R regression vector, $\boldsymbol{\beta}_{\text{PLS}}$, picked up much noise in the data. We further see the unregularised PLS-R vector, $\widehat{\boldsymbol{\beta}}_{\text{PLS}}$, found using the unregularised Krylov subspace formulation, and that it is almost indistinguishable from the regular PLS-R regression vector (the green and blue curves in Figure 1 (b)). In fact, the differences are attributed to numerical instability in the noise dimensions, because the differences disappeared for few components, and when the number of components were near the rank of the data matrix. The regression vector found when using the unregularised Krylov subspace formulation had much less noise, and had seven variables (about 7 %) that were smaller than $5 \cdot 10^{-7}$ (considered as zero). Note that the regression vector for the PLS-R model computed using the NIPALS algorithm did not have any coefficients that were near zero.

The found and reconstructed weight and score vectors of the models are illustrated in Figure 1 (c) and Figure 1 (d), respectively. They are all highly correlated, implying that it would be possible to interpret them in a similar way.

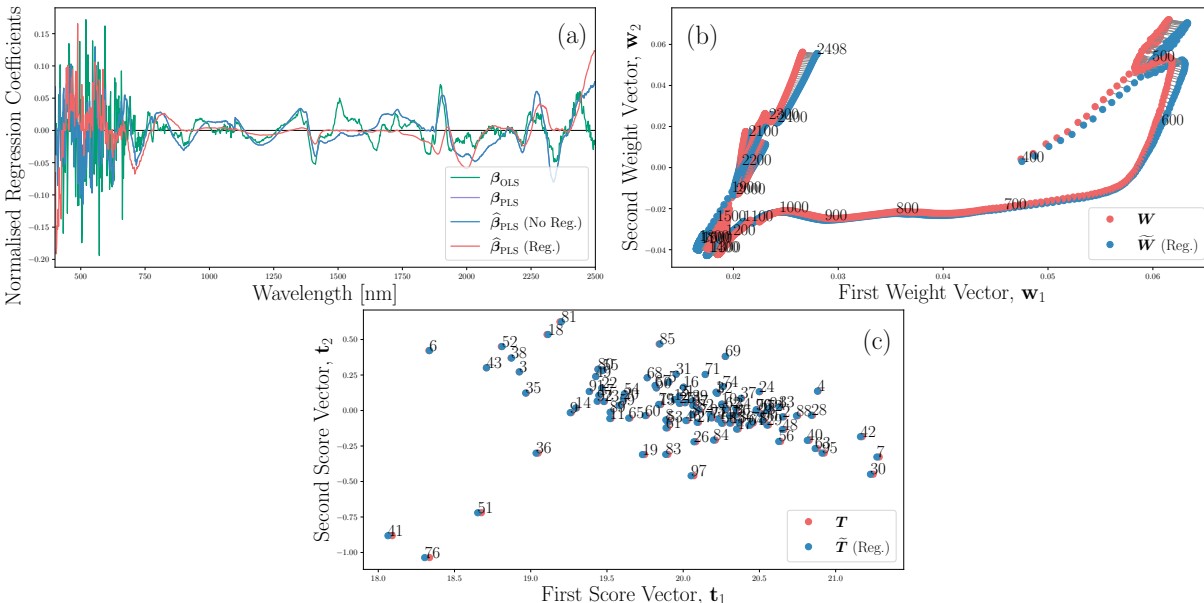

Figure 2: (a) The regression vectors from the ordinary least squares model, $\boldsymbol{\beta}_{\text{OLS}}$, the PLS-R model computed using the NIPALS algorithm, $\boldsymbol{\beta}_{\text{PLS}}$, the PLS-R model computed using the Krylov formulation without regularisation, $\widehat{\boldsymbol{\beta}}_{\text{PLS}}$ (No Reg.), and the PLS-R model computed using the Krylov formulation with elastic net regularisation, $\widehat{\boldsymbol{\beta}}_{\text{PLS}}$ (Reg.). Note that the purple and blue lines are indistinguishable. (b) The first and second weight vectors for the PLS-R model computed using the NIPALS algorithm (red curve), and for the PLS-R model computed using the Krylov formulation with elastic net regularisation (blue curve). The numbers indicate the wavelengths (in nm). (c) The first and second score vectors for the PLS-R model computed using the NIPALS algorithm (red points), and for the PLS-R model computed using the Krylov formulation with elastic net regularisation (blue points). The numbers indicate the sample index.

## 3.2 Example 2: Soil Samples Measured with NIR

The second example contains soil samples originating from a long-term field experiment in Abisko, Sweden, described by Rinnan & Rinnan (2007)[1]. Each of 36 samples were collected from the 5 to 10 cm depth

---

[1]Obtained from http://www.models.life.ku.dk/NIRsoil.

with three repetitions, yielding a total of $n = 108$ samples. The samples were scanned using NIR, in the wavelength range of 400–2498 nm at $p = 1050$ wavelengths. The target variable was soil organic matter (SOM, *e.g.* plant residues), that was measured as loss on ignition at 550 °C.

Again, we fit one regular PLS-R model and one PLS-R model based on the Krylov subspace formulation with elastic net regularisation. The number of components extracted were $K = 13$ for the regular PLS-R model, and it was $K = 50$ with $\gamma = 1.0 \cdot 10^{-5}$ and $\lambda = 1.0 \cdot 10^{-3}$ for the elastic net regularised PLS-R model based on the Krylov formulation. We also fit an ordinary least squares (OLS) regression model, and one PLS-R model using the Krylov formulation without the elastic net regularisation.

The PLS-R regression vectors are illustrated in Figure 2 (a). We see that the OLS vector is very noisy (green curve), and that the PLS-R regression vectors are less so (purple and blue curves). Further, we see that the regression vector from the regularised PLS-R model (red curve) has many values close to zero, especially in the range of about 900–1750 nm. The regression coefficient values are close to zero, and it has a sparsity strucure with 27 coefficients being zero (smaller than $5 \cdot 10^{-7}$), or about 2.5 %. Note that the regression vector for the PLS-R model computed using the NIPALS algorithm did not have any coefficients that were near zero.

The reconstructed first and second weight and score vectors, from the regularised PLS-R model, are illustrated in Figure 2 (b) and (c), respectively, together with the weight and score vectors from the PLS-R model computed using the NIPALS algorithm. We see that they are very close, implying that it would be possible to interpret them in a similar way.

## 4 Discussion and Conclusions

We have presented a simple way to use the Krylov formulation to solve the PLS-R problem, which allows additional regularisation terms to be added to the model. We illustrated the use of elastic net regularisation ($\ell_1$ and squared $\ell_2$ terms) for additional regularisation and variable selection, and demonstrated that the found regression vectors were sparse.

Note, however, that while we illustrated that the proposed formulation allow sparse regression vectors, the proposed formulation allows an analyst to impose any conceivable problem-relevant penalties in the PLS-R model. Further, using the Krylov formulation allows other solvers, for instance more efficient Krylov-based solvers to be used for the PLS-R problem.

Further, we proposed an approach to approximate the scores and loadings for the PLS-R regression vector found using the Krylov formulation, which allows interpretations of the model in the same way as PLS-R models are interpreted when they are computed using *e.g.* the NIPALS algorithm.

We illustrated the utility of the model on simulated data, and on a real data set with soil sample data. Both examples showed that the Krylov PLS-R method gave regression coefficient vectors that had coefficients that were zero or close to zero, meaning that variable selection was performed. Further, both examples showed that it was possible to approximate weight and score vectors for the Krylov PLS-R model, that were close to the PLS-R equivalents, and that thus can be used for model and data interpretation.

The proposed PLS-R model formulation opens the door to more elaborate regularisation in a PLS-R model, while still allowing corresponding scores, weights, and loadings to be approximated. Follow-up research could also focus on computational aspects, to *e.g.* speed up the computations required for the component reconstructions.

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
