# OpenReview forum: "Using the Krylov Subspace Formulation to Improve Regularisation and Interpretation in Partial Least Squares Regression"
_TMLR — Rejected by TMLR_

### Review · Reviewer_eEF6 · 2022-07-12

**Summary Of Contributions:**

This paper studies the classical partial least square regression (PLS-R) problem and its regularization version. This paper proposes an alternative way of expressing the regularized PLS-R problem as an constrained minimization problem and introduces an ADMM type approach to solve it.


**Broader Impact Concerns:**

I do not see any concerns on the broader impact.

**Requested Changes:**

1. Lemma 1.
The presentation of Lemma is strange and needs some revision.
- First, the statement requires \emph{If ${\mathcal{K}_K(...)}$ is a basis for weight vectors, ...}.
	But in the first sentence of the proof, the authors directly argue that this is a known result.
	If this is a known result, it should not be casted in the `if' statement.
- Avoid using the same notations. The notation $\mathbf{A}, \mathbf{v}$ already appear in Definition 1.
	So they should be replaced by other notations to avoid confusion.

2. An explicit example of the matrix $\mathbf{K}$.
Since the matrix $\mathbf{K}$ appears very frequently in the paper but it was not explicitly defined,
I would recommend the authors to at least provide a clear way to construct it.
For instance, one can first form a matrix whose columns are $X^Ty, (X^TX)^{-1}X^Ty, ...$
and then perform an orthonormalization.

3. Section 2.3.1. Some clarifications are needed to make this part more clear.
- Is $\mathbf{\alpha} = \mathbf{x}$?
-  The authors should add some comments on the meaning of $\mathbf{x}, \mathbf{z}$ and how they are presented in a usual ADMM algorithm.
- Running the algorithm requires an initial choice of $\mathbf{x}^{(0)}, \mathbf{z}^{(0)}$. This should be clarified.
- Will the optimization from this part always be the same as the output of equation (4)?
	If so, the authors should provide a proof.
	If not, then the authors should argue why we want to do it.

4. Motivation of Section 2.4.
	I was confused why we need to match the result from equation (4) to equation (2).
	It is known that the regularized solution will generally not be the same as the unregularized solution.

	Why do we need this procedure?

	A common motivation for regularization is the high-dimensional scenario where the unregularized problem cannot be solved.
	But if this is the case, the equation (2) is not solvable so the method in Section 2.4 cannot be used.

5.  Why will the solution of  $\hat \beta_{PLS,k}$ be different from $\hat \beta_{PLS}$?
	It seems to me that a simple solution to equation (7-16) is just setting all tilde versions to be the same as the non-tilde version,
	which eventually leads to $\hat \beta_{PLS}=\hat \beta_{PLS,k}$ when $k=K$.

	Why this is not a feasible solution?

	Is the problem comes from $k\neq K$?
	However, $k$ has to be smaller than or equal to $K$ otherwise the formulation is ill-defined.
	In the case $k\leq K$, the solution seems to be simply the PLS-R with $k$ basis rather than $K$ basis.

	Also, how does this related to the problem in Section 2.3?

6. The definition of $\tilde{{W}}$,
$\tilde{{P}}$,
$\tilde{{C}}$.

	The definition of $\tilde W_k$ can be simply written as $\tilde W_{k+1} = [\tilde w_{1},\cdots, \tilde w_{k+1}]$, where each $\tilde w_{j} = \mathbf{K} \tilde \omega_{j}$.
	The presentation in page 7, line 4 makes it more involved.

	Also, it should be clarified that $\tilde P_{k+1}, \tilde C_{k+1}$ consist of column vectors of $\tilde p_{k}, \tilde c_{j}$. But these column vectors are defined later.
	These column vectors should be defined along with these matrices.

7.  Definition of $\hat \beta_{PLS,k}$.
	This vector was used and stated in equation (29)
	but its formal definition is given later in Section 2.4.4.
	Please define this vector when it first appears.

8. Theorem 1.
	Why this result is stated as a theorem rather than a lemma (compared with Lemma 1)?
	A theorem is generally a more important and useful result than a lemma.

	What does Theorem 1 help us?
	It only shows that the solution lies in the Krylov space but it says nothing about its optimality so its prediction power could be very bad.

9. Choice of the step size $\eta$.
	The step size always plays a key role in a gradient decent algorithm.
	The authors should provide some discussions on its choice and even some analysis on
	the feasible range of it.



**Strengths And Weaknesses:**

Strength:
- This paper considers a alternative framework of expressing the partial least square
- This paper shows how this alternative expression can be easily combined with regularization

Weakness:
- The writing of this paper can be improved a lot
- The motivation of some extensions in this paper was not very clear
- There is a lack of analysis on the proposed method

---

> ### Author Response · Authors · 2022-08-02
> **Response to Reviewer eEF6 on Paper200**
>
> Thank you for carefully reading the paper and for the constructive comments and suggestions. We have incorporated a large part of the proposed changes. We go through the points raised below, and how we addressed them or our motivation should we disagree.
>
>
> **1. Lemma 1.**
>
>  * Point 1: We agree, it was awkward to state that in the proof. We have removed it, to avoid any confusion and to make the point more clear.
>
>  * Point 2: We have changed from $\mathbf{A},\mathbf{v}$ to $\mathbf{M},\mathbf{u}$ to avoid confusion.
>
>
> **2. Example of matrix $\mathbf{K}$**
>
> We have now added a concrete example directly after Definition 1.
>
>
> **3. Section 2.3.1. Clarifications.**
>
>  * Point 1: Thank you, this was an error. Where it says $\mathbf{x}$ it should say $\boldsymbol{\alpha}$. Corrected now.
>
>  * Point 2: We agree this could be useful to some readers, but we feel that explaining the details of ADMM is outside of the scope of the paper. It is clear from Equation 5 that if the constraint is active, the formulations are equivalent and $\mathbf{z} = \mathbf{K}\boldsymbol{\alpha}$. The provided references, Gabay and Mercier (1976) and Boyd *et al.* (2010), should be enough to lead the interested reader in the right direction.
>
>  * Point 3: We now mention the starting values after the steps of the ADMM algorithm are presented. We initialised $\boldsymbol{\alpha}^{(0)}$ to random independent standard Gaussian values, and let $\mathbf{z}^{(0)}=\boldsymbol{\upsilon}^{(0)}=\mathbf{K}\boldsymbol{\alpha}^{(0)}$.
>
>  * Point 4: The problems are equivalent. There are many convergence results for ADMM in the literature. Equation 6 is convex (and under some mild assumptions on $L_\rho$), it will converge to the optimal value. See the references for more details. We believe that ADMM well-known, and we shouldn't need to provide a convergence proof. That would be outside of the scope, and can be found elsewhere. We added a note about convergence, and that at convergence ($s\rightarrow\infty$), we have $\mathbf{K}\boldsymbol{\alpha}^{(s)}-\mathbf{z}^{(s)}\rightarrow \mathbf{0}$, and thus that the found $\boldsymbol{\alpha}^{(s)}$ is also a solution to Equation 4.
>
>
> **4. Section 2.4. Motivation.**
>
> We agree on the motivation of the second step. We have clarified this first paragraph in Section 2.4. With the Krylov formulation, we have a regression vector that we can use for predictions, but no longer have the model components (weights, scores, and loadings from Equation 1), and can no longer interpret the model as in traditional PLS-R. We therefore propose a way to compute approximate components, with the same properties as the original PLS-R components, which gives a regression vector ``close'' to the regression vector from the Krylov formulation. We have clarified this in Section 2.4, and hope it is more clear why we need this second procedure.
>
>
> **5. Why will the solution of $\hat{\beta}_{PLS,k}$ be different from $\hat{\beta}\_{PLS}$?**
>
> This is main trade-off in Equations 7--16: If the constraint parameters in Equations 12--15 be zero, we recover the components from Equation 1, but then the constraint in Equation 16 is very large. If the constraint in Equation 16 be zero, then the approximated model components deviate much from the original PLS-R model components in Equations 12--15. We can only get model components that reconstruct the regression vector well (Equation 16 be small) or that reconstruct the model components well (Equations 12--15 be small). We now explain this at the end of Section 2.4.
>
> The number of components, $k$: Note that $\hat{\boldsymbol{\beta}}_{PLS,k}$ is the regression vector constructed from the approximated model components using Equation 2, while $\hat{\boldsymbol{\beta}}\_{PLS}$ is the regression vector found using the regularised Krylov subspace formulation in Equation 4. These only coincide when Equation 16 be small and Equations 12--15 be large, as explained above.
>
>
> **6. The definition of $\tilde{W}$, $\tilde{P}$, $\tilde{C}$.**
>
> We have clarified this, and have removed the definition from there, and have instead put the definitions of $\mathbf{W}_k$, $\mathbf{T}_k$, $\mathbf{P}_k$, and $\mathbf{C}_k$ directly after their first use in Equations 9--11 and Equation 16. We hope this makes it more clear.
>
>
> **7. Definition of $\hat{\beta}_{PLS,k}$.**
>
> We now state the definition clearly, and put it directly above Equation 30, where it is first used.
>
>
> **8. Theorem 1.**
>
> We have now *downgraded* the Theorem to a Lemma (2). The result in Lemma 2 is not grande, but may prove useful in the future if new algorithms are devised to solve the program in Equation 17.
>
>
> **9. The step size.**
>
> The initial step size was set to 0.0001. In each iteration, if $f$ decreased, then $\eta^{(s+1)} = \eta^{(s)}$ and if $f$ increased, the step size was divided by a factor $1 + 1 / (1 + 0.01s)$, such that $\eta^{(s+1)} = \eta^{(s)} / (1 + 1 / (1 + 0.01s))$. We have added a description of this step size heuristic to Section 2.4.3.

---

### Review · Reviewer_WZQT · 2022-07-22

**Summary Of Contributions:**

This paper proposes a regularization framework for partial least squares regression by exploiting a Krylov subspace formulation of the problem and adding an elastic net penalty on the regression coefficients. Furthermore, the paper formulates an optimization problem for reconstructing the weights, and hence the scores and loadings, from the estimated regression coefficients. The paper derives an ADMM algorithm for solving the former problem and a projected gradient descent algorithm for solving the latter. The proposed method is illustrated on simulated and real data.

Overall, the paper addresses an interesting problem, and the proposed method is potentially useful for applications where simultaneous dimension reduction and variable selection are desired. However, the proposed optimization algorithms are standard, and no statistical or algorithmic properties are presented. Also, the practical issues of tuning parameter selection are not discussed. Therefore, the contributions of the paper seem fairly limited.

**Broader Impact Concerns:**

None.

**Requested Changes:**

1) It would be interesting to understand how the proposed method performs in high-dimensional settings, in view of the recent theoretical developments in PLS regression (e.g., Cook and Forzani, 2019, “Partial least squares prediction in high-dimensional regression,” *Ann. Statist.*, 47, 884-908).
2) The convergence properties of the projected gradient descent algorithm need to be better understood.
3) Is it possible to compute the $K$ vectors of weights simultaneously?
4) Why are the weights, scores, and loadings shrunk toward their unregularized counterparts rather than toward zeros? What if the regularized regression coefficients are far away from the unregularized version?
5) Tuning parameter selection is an important part of the regularization problem, and should be carefully discussed. Cross-validation is usually a safe option, but would be too expensive for optimizing so many tuning parameters. Is it possible to derive an information criterion, such as BIC, in this case? To this end, an estimate of the degrees of freedom may be needed; see, e.g., Krämer and Sugiyama (2011, “The degrees of freedom of partial least squares regression,” *JASA*, 106, 697-705).

**Strengths And Weaknesses:**

Strengths:
1) The Krylov subspace formulation is less explored for computing purposes in the PLS regression literature. The idea of directly penalizing the regression coefficients in this formulation seems novel.
2) The proposed elastic net regularized problem leads to sparse regression coefficients, and hence performs variable selection and improves the interpretability of the PLS method.

Weaknesses:
1) Given the basis $\mathbf{K}$, problem (4) is just a generalized elastic net problem, or equivalently a subspace constrained elastic net problem, which has been studied by, e.g., Mouret, Brault, and Vahid Partovinia (2013, “Generalized elastic net regression,” *JSM Proceedings*, 3457-3464). With the PLS regression and Krylov subspace structures taken into account, of course, the statistical properties of the proposed estimators may not be trivial. However, the paper did not go further in this direction.
2) The proposed optimization problem for reconstructing the weights is nonconvex, and its convergence properties with projected gradient descent is unclear. The term (19) in the objective function is highly nonconvex with the weights appearing in the matrix inverse, which may cause instability.
3) One major advantage of the Krylov subspace formulation is that the $K$ vectors of regression coefficients can be obtained simultaneously. This important advantage, however, is lost in the reconstruction algorithm, which finds the $K$ vectors of weights in a sequential manner.
4) The tuning parameters $\gamma$ and $\lambda$ are critical to the performance of the elastic net regularized estimator. In the simulated and real data examples, these parameters seem to be arbitrarily specified, resulting in a very low level of sparsity (7% and 2.5%, respectively). With such low sparsity levels, the elastic net penalty does not essentially play a role, and the trade-off between data fitting and sparsity is not well illustrated.
5) The optimization problem for reconstruction involves five tuning parameters, which are too many and could be very difficult to choose in practice. The paper did not even mention how these parameters were specified in the data examples.

---

> ### Author Response · Authors · 2022-08-07
> **Response to Reviewer WZQT on Paper200**
>
> Thank you for carefully reading the paper and your constructive comments and suggestions. We have updated the manuscript and addressed the comments. We go through the comments below, and explain how we have addressed them.
>
> **Generalized elastic net regression**
>
> The program in Equation 4 is similar to generalized elastic net regression (GENR), but is distinct. In the notation of GENR, the proposed problem has $\mathbf{D}_1=\mathbf{D}_2=\mathbf{K}$, but the design matrix in GENR would be $\mathbf{X}\mathbf{K}$, and $\mathbf{K}$ depends on both $\mathbf{X}$ and $\mathbf{y}$, so is not a prior. In the Bayesian interpretation it is part of the likelihood.
>
> **Instability**
>
> It would be possible to include penalties like Equation 19 directly in Equation 1, but requires one such equation for every penalty (2 here). A much more difficult problem, and more unstable. The Krylov formulation allows restricting the formulation in Equation 19 to a single occurrence, one of the benefits of the proposed approach. Instability would happen for components with large $k$ when the $\mathbf{t}_k$ norm is very small (see Höskuldsson, 2003), and should be handled when determining the number of components.
>
> **Tuning parameters**
>
> We used 10-fold cross-validation (CV) to find $\gamma$ and $\lambda$, based on being better or not significantly worse than the non-regularised formulation (*wrt.* average coefficient of determination, $R^2$). We have clarified this now.
>
> **High-dimensional settings**
>
> The asymptotic properties of PLS-R with elastic net is an interesting direction. Those developments used the algorithmic formulation of PLS-R; and note that it would be easier to use a principled optimisation formulation, like we propose here. Our work illustrates how to add arbitrary penalties to the PLS-R model, and find model components for the Krylov formulation. We therefore feel that this would fall outside of the scope of the present work. This is however very interesting for our future work.
>
> **Convergence properties**
>
> To converge to a stationary point, projected gradient descent requires the function to be continuously differentiable with Lipschitz-continuous gradient and a convex constraint set. Equation 17 is continuously differentiable: Since we assume that $k\leq \mathrm{rank}(\mathbf{X})$, then $\widetilde{\boldsymbol{\omega}}^T\mathbf{K}^T\mathbf{X}^T\mathbf{X}\mathbf{K}\widetilde{\boldsymbol{\omega}}>0$. Hence, Equations 18-23 are all non-trivial. Equation 29 is differentiable and non-degenerate, since $\widetilde{\boldsymbol{\omega}}^T\mathbf{B}\widetilde{\boldsymbol{\omega}}>0$, and the rest are just linear or quadratic.
>
> Equation 17 is continuously differentiable, the multivariate mean value theorem says there is a Lipschitz constant (in the 2-norm), and that it is the spectral norm of the Jacobian of the gradient function of Equation 17.
>
> We converge with a constant step size smaller than 2 over the Lipschitz constant. However, we used the heuristic: Initial step size 0.0001 (trial and error). In each iteration, if $f$ decreased, then $\eta^{(s+1)}=\eta^{(s)}$; if $f$ increased, then $\eta^{(s+1)}=\eta^{(s)}/(1+1/(1+0.01s))$. We now describe this heuristic in Section~2.4.3.
>
> The feasible set is convex, and the projection computed to full numerical precision, so no need to take special care of the projection.
>
> **Compute weights simultaneously?**
>
> Yes, but simultaneous computation is non-trivial, because of the relationships between weights, and scores and loadings. With one component, the score vector is $\mathbf{t}=\mathbf{X}\mathbf{w}$, but we can't do $\mathbf{T}$ as $\mathbf{T}=\mathbf{X}\mathbf{W}$. Instead we do $\mathbf{T}=\mathbf{X}\mathbf{W}(\mathbf{P}^T\mathbf{W})^{-1}$ (see Höskuldsson, 2003). More complicated for the scores, and even more for the loadings. The simultaneous problem is *much* more difficult.
>
> **Shrinkage of weights, scores, and loadings**
>
> Good point, we have now clarified this. Little regularisation gives similar components as traditional PLS-R, but may deviate if much regularisation. This is the main trade-off in Equations 7-16: If the parameters in Equations 12-15 be zero, recover the components from Equation 1, but then Equation 16 is large. If the constraint in Equation 16 be zero, the approximated components deviate from original PLS-R components (Equations 12-15). Model components reconstruct the regression vector well (Equation 16 small) or the model components well (Equations 12-15 small). We now explain this better in Section 2.4.
>
> **Selecting tuning parameters**
>
> This is important, and tuning becomes difficult with many parameters. We used trial and error in the examples (large regression parameter worked well), but principled approaches are warranted. An information criterion might be relevant, but would need careful consideration and principled tests. Outside of the scope of the present study, but should be the subject of future studies. We added a note to the discussion section.

---

> > ### Comment · Reviewer_WZQT · 2022-08-09
> > **Thanks, but a long way to go**
> >
> > 1) Similarity to generalized elastic net: I certainly note that $\mathbf{K}$ depends on the data and is not a prior, but this matters only when statistical properties are considered.
> > 2) Trial and error is usually not an acceptable strategy for parameter tuning, since it leaves too much room for cherry-picking.
> >
> > My main point is that the contributions of the present work are not sufficient to justify its publication in a decent journal. I would expect the authors to address at least some nontrivial aspects in theory or methodology as mentioned in my review and improve the work substantially.

---

> > > ### Author Response · Authors · 2022-08-10
> > > **Properties and hyperparameters**
> > >
> > > We would be interested in looking at the shrinkage properties (and if their peculiar pattern improves) as we add other penalties to it. Would you be interested in such an addition to the paper?
> > >
> > > Just to be clear: We are not trying to fool anyone here by cherry-picking, or other means. The reconstruction problem seem to have a rather small set of feasible parameters, and in fact we didn't obtain any solutions unless the weight parameter was rather large. This suggests the found regression vector is "far" (in some sense) from the one found through traditional PLS-R. However, we could do this search by for instance Bayesian optimisation instead, and look for as small a difference in the regression vector under the condition of a feasible solution. Would you find that acceptable?

---

### Review · Reviewer_tFDm · 2022-07-25

**Summary Of Contributions:**

This paper studies the Partial Least Squares Regression (PLSR) method, a method for linear regression with dimensionality reduction. The main contribution of the paper is to propose a regularized version of this method utilizing a Krylov subspace reformulation, along with an approximate optimization method for the regularized problem.

**Broader Impact Concerns:**

I don’t see broader impact or ethical concerns regarding this paper.

**Requested Changes:**

Please find my questions in the Strengths and Weaknesses part.

**Strengths And Weaknesses:**

My overall feeling is that the topic of the paper may not be of wide interest to the community, and the contribution of the paper on this topic is quite questionable as well.

The main contributions on the theory / methodology side seem to be in two sections, Section 2.3 and 2.4. Section 2.3 utilizes the known Krylov subspace reformulation to incorporate a regularization into the problem, turning Eq (3) to Eq (5), and deriving an optimization procedure using ADMM. All these seem to be standard well-known practice. Then I have a hard time understanding what is the purpose of Section 2.4—Is it some approximate optimization method? If so then what is the advantage of the approximation? (Faster runtime, or some reformulation into some standard problem or software?) In contrast, what is the issue with the ADMM method in Section 2.3.1?

For the experiments, I also don’t understand what the conclusions are. Upon reading Section 3 it feels like main claims are that the proposed optimization via the reformulation successfully reproduces the existing implementation (for the unregularized case), and then can also produce the regularized case. This feels quite standard and does not sound like a novel research result. Also, the figures seem mostly like basic visualizations of the learned parameters, and does not report more concrete comparisons of quantitative metrics, such as train / test errors, or runtime.

I also think the topic of the paper (PLSR method) sounds quite nuanced, and I am not sure how interesting it is to the broader ML/Stats community (at least from reading the paper). As the authors mentioned, the method is widely used in certain application areas in science, which may be a better audience for the present paper (where the method is considered standard).

As for a general ML/stats method, the best connection I can draw is its similarity to Principal Components Regression (PCR), as the authors mentioned, which is another classical regression method based on data-dependent dimensionality reduction. However, I don’t think either PCR or PLSR are as widely used or considered, at least as general methodologies, compared with e.g. Lasso, ridge regression, or elastic net. The theoretical advantages of PLSR are unclear (the weightings depend on both X and y, unlike PCR). The practical advantages such as interpretability are only briefly mentioned in the paper, and without further explanations and/or corresponding results.

---

> ### Author Response · Authors · 2022-08-04
> **Response to Reviewer tFDm on Paper200**
>
> Thank you for carefully reading the paper and the constructive comments and suggestions you have made. We are working on an updated version of the manuscript that incorporates the proposed changes. We go through the points raised below, and explain how we have addressed them or our motivation should we disagree.
>
> **The purpose of the reconstruction in Section 2.4.**
>
> The benefit of using the Krylov formulation is that we can add arbitrary penalties to the PLS-R problem in this case. We exemplify this by adding elastic net regularisation, and devise the steps of ADMM to solve the resulting optimisation problem.
>
> The PLS-R problem, as typically formulated (Equation 1), implicitly determines the regression coefficients through the model components in the dimensionality reduction (Equation 2).
>
> Now, if we instead of Equation 1 use the Krylov formulation, we directly solve for the regression coefficient vector instead of for the model components, and we therefore do not have any model components. The dimensionality reduction is now instead done implicitly through the Krylov matrix, $\mathbf{K}$. One of the main benefits that applied researchers see in the PLS-R method is the ability to analyse the model components. They can for instance reveal which variables (columns of $\mathbf{X}$) drive which samples (rows of $\mathbf{X}$) towards a certain prediction, and overall give many different ways to interpret both the data and the model predictions.
>
> What we propose in this paper, is a way to estimate model components (weights, scores, and loadings) that we *would have gotten*, had we obtained a particular regression vector. We try to find such models component vectors that give us a regression vector that is *close* to the regression vector that we obtained through the Krylov formulation of the problem. In the case when we don't have any added regularisation (when we don't have the elastic net penalty) we obtain *exactly the same* model components that we got through the traditional PLS-R formulation (using Equation 1, and we illustrate this in the examples), but if we add regularisation terms (when we do add the elastic net penalty to the Krylov formulation of the problem), then we will (in the general case) only obtain model components that approximate the regression vector we found using the Krylov formulation. We show in the examples that the model components that we find are indeed *similar* (correlated) to the ones we obtain through traditional PLS-R, and therefore we conclude that they can be used for interpretation as well, but of course if the caveat that they are only approximated.
>
> The advantage is then that model components for interpretation are now (thanks to our proposed approximation method) available also when using the Krylov formulation of the PLS-R problem. The lack of model components is likely why the Krylov formulation has not been used for PLS-R. We have tried to make this more clear in the paper and in the conclusion section.
>
>
> **Conclusions of the experiments.**
>
> The main novelty that we propose is a method to approximate model components in the Krylov formulation, where they are not available.
>
> The examples show that: 1) in the unregularised case (without elastic net), the results are indeed the same as that from the traditional formulation of PLS-R (Equation 1), hence, *the approximation method works as stated*; and 2) in the regularised case (with elastic net regularisation added), we obtain model components that reveal similar relationships between scores and loadings as in the unregularised case (they can be interpreted in a similar way) but that also give rise to a sparse regression vector.
>
> The point is not to show quantitative improvements in prediction performance, but to show that with the proposed developments it is possible to: 1) obtain a sparse regression vector and 2) to obtain model components that are interpretable.
>
> To make the interest in the interpretation more clear, we have added more information about the interpretation of PLS-R models.
>
>
> **Interest in the PLS-R method itself**
>
> The lack of interest in PLS-R is perhaps a subjective assessment, but we do understand the sentiment. PLS-R has not been widely used in machine learning, but we would argue that much work with these methods have been done in the statistics community.
>
> Note for instance that PLS-R is implemented in scikit-learn. Note also that up until today there are 17,900 papers published only in 2022 that mention ``partial least squares'' (see for instance https://scholar.google.se/scholar?as_ylo=2022&q=%22partial+least+squares%22).
>
> There is thus substantial interest in such methods, although, granted, most of it is in applied science fields. On the other hand, the present work is not applied, and while not heavy on the theoretical side, it clearly falls under machine learning method development.

---

### Review · Reviewer_Dqev · 2022-08-01

**Summary Of Contributions:**

Devised an algorithm to solve the (regularized) partial least squares (PLS) regression problem using the Krylov subspace formulation, where the objective is to minimize the squared error (on the subspace bases) plus an elastic net penalty ($\ell_1$ + $\ell_2$ regularization). The objective is solved via ADMM, after which a second optimization problem is introduced to construct parameters (scores and loadings) that remain close to the regularized solution $\hat{\boldsymbol{\beta}}_{PLS}$ and also satisfy the PLS constraint. The algorithm is applied to solve several real-world regression problems.

**Requested Changes:**

Please refer to the weakness section above.

**Strengths And Weaknesses:**

I am not familiar with the PLS literature so I cannot evaluate the novelty and significance of the proposed method.
In my opinion the submission does a good job introducing the problem setting (PLS, Krylov subspace, etc.) and explaining the optimization objective (relaxation/approximation, etc.). Experiments also illustrate the effectiveness of the added regularizations.

I have the following concerns & questions.

1. After reading the main text, I'm still not sure if I understand the motivation of using PLS, which involves a rather opaque formulation. As mentioned by the authors in the introduction, the main advantage seems to be that dimension reduction is performed by taking the target information into account. If this is the case, then some empirical or theoretical comparison against alternative methods (such as PCR) would strengthen the manuscript.

2. Related to the previous point, for PCR it is well-known that certain notion of "alignment" between the target and the data covariance affects the prediction error (for example see [Huang et al. 2020] and reference therein). It would be nice to at least empirically evaluate the performance of the proposed method on synthetic datasets, where the ground truth and alignment conditions can be manipulated.
Huang et al. 2020. Dimensionality reduction, regularization, and generalization in overparameterized regressions.

3. A large part of the main text is devoted to constructing the score/loading parameters from the approximate solution $\hat{\boldsymbol{\beta}}_{PLS}$. How does this compare to directly solving for the minimum $\ell_1 + \ell_2$ norm solution, instead of using explicit regularizations as in Equation (4)?

4. The time and space complexity of the proposed method is not analyzed. I suspect that due to non-convexity of the formulation, global optimization guarantee is probably difficult to obtain; but such limitation should be discussed more explicitly.

5. (minor) Please double-check the typos. For example, wouldn't the equality constraint after Equation (5) be $\boldsymbol{K\alpha}=\boldsymbol{z}$ instead?

---

> ### Author Response · Authors · 2022-08-07
> **Response to Reviewer Dqev on Paper200**
>
> Thank you for carefully reading the paper and the constructive comments and suggestions you have made. We are working on an updated version of the manuscript that incorporates the proposed changes. We go through the points raised below, and explain how we have addressed them or our motivation should we disagree.
>
> **Motivation for using PLS-R**
>
> We agree that the opaque nature of PLS-R is a problem. This is in part the motivation for us to develop a more principled formulation by using the Krylov subspace approach. It would be easier for practitioners to motivate the use of PLS-R if it has a formulation that is easier to understand and to modify (for instance to add new regularisation terms). The problem has been that practitioners want to use the PLS-R model components to understand and interpret their models, and this was previously not possible when using the Krylov formulation. This is what we have tried to resolve in this work, to provide (approximate) model components for the Krylov formulation of the PLS-R problem.
>
> There are some areas where PLS-R has been (practically) shown to perform much better than other regression methods, and one such application is when analysing spectral data (hence the example in the manuscript). We do not question the use of PLS-R *per se* in this work, but instead want to provide a more principled approach for those who (for whatever reasons) want to use PLS-R to analyse their data. In that case, the Krylov formulation is a better choice, since it is principled, and since it allows other regularisation terms to be included. (It may also prove in the future to lead to more efficient algorithms, since it allows the use of any tool from numerical linear algebra.) For that case, the Krylov formulation with penalties, we provide a means to approximate model components corresponding to the found regression vector.
>
> **Directly penalising the traditional PLS-R problem**
>
> This is a good question, and the reason we find this approach intractable is the following: We can include penalties like $\ell_1$ and $\ell_2$ directly in Equation 1, by using Equation 19. But this then requires one penalty similar to Equation 19 for each penalty we would like to add (two in the manuscript's example). For instance, we would have the $\ell_2$ penalty as,
> $\|\|\widetilde{\mathbf{W}}\_{k+1}(\widetilde{\mathbf{P}}\_{k+1}^T\widetilde{\mathbf{W}}\_{k+1})^{-1} \widetilde{\mathbf{C}}\_{k+1}^T\|\|\_2^2$,
> which would be similar to the penalty in Equation 19, but for more elaborate penalties this becomes prohibitive. In fact, we run into problems already with the $\ell_1$ penalty,
> $\|\|\widetilde{\mathbf{W}}\_{k+1}(\widetilde{\mathbf{P}}\_{k+1}^T\widetilde{\mathbf{W}}\_{k+1})^{-1} \widetilde{\mathbf{C}}\_{k+1}^T\|\|\_1$,
> which we cannot solve easily since it is non-differentiable and non-convex. And, as said above, with other and more elaborate penalties this would be even more difficult to solve, since the problem becomes highly non-linear.
>
> Going through the Krylov formulation instead, we are able to restrict the formulation in Equation 19 to a single occurrence. This is one of the benefits we see when using the proposed approach as opposed to trying to add penalties to the traditional PLS-R formulation in Equation 1.
>
> **The time and space complexity of the proposed method**
>
> The function in Equation 17 is continuously differentiable and has a Lipschitz-continuous gradient. Note that we assume that $k\leq \mathrm{rank}(\mathbf{X})$ and thus that $\widetilde{\boldsymbol{\omega}}^T\mathbf{K}^T\mathbf{X}^T\mathbf{X}\mathbf{K}\widetilde{\boldsymbol{\omega}} > 0$, and hence all terms in Equations 18--23 are non-trivial. Further, Equation 29 is differentiable and non-degenerate, since $\widetilde{\boldsymbol{\omega}}^T\mathbf{B}\widetilde{\boldsymbol{\omega}} = \widetilde{\boldsymbol{\omega}}^T\mathbf{K}^T\mathbf{X}^T\mathbf{X}\mathbf{K}\widetilde{\boldsymbol{\omega}} > 0$, and the other parts are just linear or quadratic in $\widetilde{\boldsymbol{\omega}}$.
>
> Convergence to a stationary point is then guaranteed for a constant step size smaller than 2 over the Lipschitz constant of the gradient. In this case, projected gradient descent converges to a stationary point at a rate of $\mathcal{O}(1/s)$, for $s$ the iteration counter.
>
> We have now explained and clarified this at the end of Section 2.4.3.
>
> **Typos**
>
> Thank you, this was an error. Where it says $\mathbf{x}$ it should say $\boldsymbol{\alpha}$. We have corrected this now, and gone through those equations to find other typos. Please let us know if you spot something else that we might have missed.

---

### Decision · Action_Editors · 2022-09-10

**Recommendation:** Reject

**Comment:**

This paper proposed a reformulation of partial least squares regression (PLS-R) using Krylov subspace formulation, studies an optimization problem for reconstructing weights and used ADMM and projected GD to solve it. The authors argued that by using the reformulation of PLS-R which is more principled, there are a few benefits such as allowing arbitrary regularizations. However, a few reviewers find that these benefits are not of interest to the ML community, and one reviewer argued that it will be more interesting to show the advantage in interesting metrics such as training/test error. Furthermore, one reviewer suggests a comparison to PCR to justify the advantage of the proposed methods, but the authors did not respond. Another reviewer pointed out the practical advantage of the studied formulation such as interpretability are only briefly mentioned, and there are no corresponding results to support it. I agree with the reviewers that the benefit of "allowing any regularization" is not of sufficient interest, and other benefits that may be of interest are not well demonstrated. Due to these reasons, I recommend rejection.